# The Environment and Climate Change Canada solid precipitation intercomparison data from Bratt's Lake and Caribou Creek, Saskatchewan

Craig D. Smith[1], Daqing Yang[2], Amber Ross[1], and Alan Barr[1]

[1]Environment and Climate Change Canada, Climate Research Division, Saskatoon, SK
[2]Environment and Climate Change Canada, Watershed Hydrology and Ecology Research Division, Victoria, BC

*Correspondence to:* Craig D. Smith (craig.smith2@canada.ca)

**Abstract.** Prior to the beginning of the World Meteorological Organization's (WMO) Solid Precipitation Inter-Comparison Experiment (SPICE, 2013-2015), two precipitation measurement intercomparison sites were established in Saskatchewan to help assess the systematic bias in the automated gauge measurement of solid precipitation and the impact of wind on the undercatch of snowfall. Caribou Creek, located in the southern Boreal forest, and Bratt's Lake, located in the southern plains, are a contribution to the international SPICE project but also to examine national and regional issues in measuring solid precipitation, including regional assessment of wind bias in precipitation gauges and windshield configurations commonly used in Canadian monitoring networks. Overlapping with WMO-SPICE, the Changing Cold Regions Network (CCRN) Special Observation and Analysis Period (SOAP) occurred from 2014 to 2015 involving other enhanced observations and cold regions research projects in the same geographical domain as the Saskatchewan SPICE sites. Following SPICE, the two Saskatchewan sites continued to collect core meteorological data (temperature, humidity, wind speed, etc.) as well as precipitation observations via several automated gauge configurations, including the WMO automated reference and the Meteorological Service of Canada's (MSC) network gauges. In addition, manual snow surveys to collect snow cover depth, density, and water equivalent were completed over the duration of the winter periods at the northern Caribou Creek site. Starting in the fall of 2013, the core intercomparison precipitation and ancillary data continued to be collected through the winter of 2017. Automated observations were obtained at a temporal resolution of 1 minute, subjected to a rigorous quality control process, and aggregated to a resolution of 30 minutes. The manual snow surveys at Caribou Creek were typically performed every second week during the SPICE field program with monthly surveys following the end of the SPICE intercomparison period. The Saskatchewan SPICE data are available at https://doi.org/10.18164/63773b5b-5529-4b1e-9150-10acb84d59f0. The data collected at the Saskatchewan SPICE sites will continue to be useful for transfer function testing, Numerical Weather Prediction and hydrological forecasting verification, ground truth for remote sensing applications, as well as providing reference precipitation measurements for other concurrent research applications in the cold regions.

## 1 Introduction

Cold region hydrology and climatology research and monitoring requires accurate measurements of solid precipitation, crucial for water resource forecasting, driving climate and hydrological models, and climate monitoring and trend analysis (Barnett et al., 2005; Gray et al., 2001; Bartlett et al., 2006; Laukkanen, 2004). The

systematic snowfall measurement biases, either via a manual observer or automated measurements, are well documented (e.g. Sevruk et al., 1991; Goodison et al., 1998) and have resulted in international intercomparison initiatives such as the World Meteorological Organization's (WMO) Solid Precipitation Measurement Intercomparison (Goodison et al., 1998) and the Solid Precipitation Inter-Comparison Experiment (SPICE; Rasmussen et al., 2012; Nitu et al., 2012). The objectives of these intercomparisons were to examine the relative systematic biases of a variety of instrument configurations and to provide solutions for adjusting and homogenizing solid precipitation data such as in Yang et al. (1998, 2005), Sevruk et al. (2009), Wolff et al. (2015) and Kochendorfer et al. ( 2017a, 2017b, 2018).

During SPICE, there were eight sites that operated at least one Double Fence Automated Reference (DFAR), including Caribou Creek and Bratt's Lake (Nitu et al., 2012). The DFAR configuration consists of a large octagonal double wind fence of the same specifications as used by the WMO Double Fence Intercomparison Reference (DFIR; Yang et al., 1993, Goodison et al., 1998) only with the DFIR manual Tretyakov precipitation gauge replaced with either a Geonor T-200B or an OTT Pluvio$^2$ automatic precipitation gauge (Fig. 1). The relative performance of the DFIR can be traced back to the intercomparison between the DFIR and a bush shielded gauge at the Valdai, Russia research station (Yang et al., 1993) where the DFIR was shown to have catch efficiencies of 94%, 92% and 90% for rain, mixed, and snow respectively as compared to the bush gauge. Yang (2014) further refined this intercomparison by adding more data and showed that the catch efficiency of the DFIR was 3% to 6% higher than previously shown. In turn, the performance of the DFAR can be related to the DFIR from historical comparisons (Smith, 2009) and comparisons during SPICE (Nitu & Roulet, 2016). Smith (2009) showed that the catch of the DFAR for dry snow (snowfall at air temperatures < -2° C) was approximately 86% of the total of the adjusted DFIR (using the Yang et al. 1993 adjustment), and approximately 93% of the unadjusted DFIR catch. Nitu & Roulet (2016) showed that intercomparisons between the DFIR and DFAR during SPICE yielded a DFAR catch efficiency of approximately 92%.

Another requirement of SPICE for sites operating a DFAR reference was the inclusion of an Alter shielded and unshielded gauge pair, either Geonor T-200B or OTT Pluvio$^2$. For SPICE, the intent was to broaden the intercomparisons amongst sites that were not able to install and operate a DFAR but had the capabilities to operate the Alter shielded and unshielded pair of gauges. The shielded and unshielded pair of Geonor T-200B gauges operated at Bratt's Lake is shown in Fig. 2.

One of the legacies of the WMO-SPICE project is the high quality precipitation and ancillary data set consisting of multiple precipitation gauge configurations (different gauge models with various measurement principles utilizing many wind shield designs), wind speed measurements at both gauge height and the standard 10 m height, air temperature, and often precipitation type observations (via optical sensors). The bulk of the international WMO-SPICE data set will be made available by the WMO once data agreements have been completed. In parallel to data

collection, SPICE also developed robust data quality control techniques that can be applied to both SPICE and post-SPICE data (Kochendorfer et al., 2017b).

The SPICE precipitation data set, besides being useful for intercomparing gauge configurations for performance assessment and data homogenization, is useful for remote sensing validation, hydrological modelling applications, and further refinement and testing of precipitation gauge transfer functions. Data collected in western Canada at the Saskatchewan SPICE sites is a contribution to the Changing Cold Regions Network (CCRN; DeBeer et al., 2016) and more specifically for the CCRN Special Operations and Analysis Period (SOAP) which was conducted from October 1, 2014 to September 30, 2015 across all of the CCRN "Water, Ecosystem, Cryosphere and Climate (WECC)" observatories, including the Saskatchewan WMO-SPICE sites.

## 2 Sites, methods, and instrumentation

Table 1 shows the location and climate details of both Bratt's Lake (XBK) and Caribou Creek (CCR). The locations of the sites are indicated on the map in Figure 3.

### 2.1 Caribou Creek

The Caribou Creek SPICE site was established in November of 2012 and was fully operational by February 2013. The site is located in the southern Boreal Forest, about 100 km northeast of Prince Albert, Saskatchewan. Harvested in 2004 and previously instrumented as part of the Boreal Ecosystem Research and Monitoring Site project (BERMS; Barr et al., 2012) and the FluxNet Canada program (Margolis et al., 2006), the site consists of a regenerating Jack Pine canopy with tree heights of about 2 to 3 m. This makes the site opportune for measuring precipitation in a bush sheltered area, similar to, but not exactly the same as, the Valdai site (Yang et al., 1993) where unsheltered gauges were compared with gauges located in the bush. The pre-existing Geonor T-200B (used for BERMS) was nearly ideally located within the well sheltered bush area and would become the site "Bush" gauge (Fig. 4a). Prior to the beginning of the SPICE intercomparison period, a clearing with dimensions of approximately 60 m x 40 m was created about 100 m from the bush gauge and a DFAR was constructed inside the clearing (Fig. 4b). Along with some other instrumentation tested for SPICE, the clearing also hosted the Alter shielded (Fig. 4c) and unshielded (not shown) Geonor T-200B.

Wind speed at CCR included in this data set was measured at 2 m above the ground in the clearing using a Gill cup wheel anemometer. Temperature was measured with a Campbell Scientific HMP45C mounted at 1.5 m above the ground inside a naturally aspirated radiation shield installed near the centre of the clearing.

During SPICE, CCR hosted an automated SWE (Snow Water Equivalent) sensor, and to facilitate testing and intercomparison of this sensor, manual snow surveys to measure SWE were performed every two weeks throughout the SPICE campaign (Smith et al., 2017). Following SPICE, the manual snow surveys continued to be performed

monthly (with the exception of the winter of 2015/2016 which had no snow surveys). The snow survey used a double sampling technique (Rovansek et al., 1993) in which 5 bulk density samples were taken using an ESC-30 snow tube sampler (Farnes et al., 1983) with a total of 50 snow depth measurements taken with a snow rod between the density samples. The snow survey transect started in the vegetated area south of the clearing and crossed the clearing into the vegetated area to the north.

## 2.2 Bratt's Lake

The Bratt's Lake observatory is located approximately 30 km southwest of Regina, Saskatchewan. The site is situated on the open prairie with very little topographic relief, resulting in high exposure and therefore relatively high wind speeds (Table 1). The observation site is mown grass surrounded by agricultural crops. The lack of vegetation other than short grasses enhances the exposure. The precipitation infrastructure was installed in 2003 and included a DFIR as the manual reference for the DFAR as well as other automatic gauges including the Alter shielded Geonor T-200B (Smith, 2009). Prior to SPICE, the site was fully automated, including the two DFARs (Figure 1 right) and the same Alter shielded and unshielded Geonor T-200B precipitation gauges (Fig. 2) as at CCR. Wind speed was measured by an R.M. Young propeller anemometer at a height of approximately 2 m above the ground. Temperature and relative humidity (not reported) were measured with a Campbell Scientific HMP45C instrument inside a fan aspirated Stevenson screen at 1.5 m above the ground. Unlike CCR, there were no manual snow surveys performed at XBK.

## 2.3 Precipitation gauge heating

Prior to the start of the SPICE field campaigns, the organizing committee decided that the reference precipitation gauges used for SPICE needed to have rim heaters to prevent gauge capping (where the gauge orifice is blocked or partially blocked with snow). For the Geonor gauges discussed here, the heaters and thermistors for monitoring and switching were added prior to installation in the field. The heaters can be seen in Fig. 2 (right) where the external "chimney" is wrapped with a heating element (seen as yellow in the photo). The heating tape extends down into the lower "chimney" which is not visible in the photo, thus preventing melted snow from refreezing in the lower chimney before reaching the storage bucket inside the gauge. The heaters were turned on when the air temperature dropped below 2 °C and were controlled by thermistors embedded in the gauge rim such that the rim temperature did not exceed 2 °C. There were no lower temperature limits to the heater switching, but it was observed that the heaters could not maintain the rim temperatures at 2 °C when the air temperature was below -5 °C, although they generally kept the rim temperature above the ambient air temperature.

## 2.4 Precipitation gauge "charging"

To prevent freezing of the precipitation gauge bucket contents over the winter, the gauges were "charged" with 3 to 4 L of an antifreeze mixture consisting of 60% methanol and 40% propylene glycol. The methanol serves to decrease the density of the antifreeze mixture so that the contents do not stratify and freeze. A lightweight electrical

insulating oil (approximately 0.5 L) was then poured on top of the bucket contents to prevent evaporation of both the antifreeze and the collected precipitation.

**3 Data collection, quality control, and post-processing**

3.1 Data collection

Since both the XBK and CCR sites were required to be consistent with the other international SPICE sites, the data collection frequency for the automated data was standardized. The data loggers performed a program execution and instrument read every 20 seconds and these data were averaged and output once per minute. The 1-minute data were stored on the site data loggers and retrieved daily by the site computer. Typically once per week, the site computers were accessed remotely and the data were retrieved for quality control and post-processing.

3.2 Quality assurance and control

Following retrieval, the 1-minute data were filed into time consistent (i.e. no gaps in the time series even if the data are missing) monthly files. The data were graphed and the time series examined for instrument failures and inconsistencies. The same quality control process applied to the international SPICE data (i.e. Kochendorfer et al., 2017b) was used for our data on both the SPICE and post-SPICE observation periods. This was an automated process which removed out-of-range outliers and data jumps, the thresholds for which were set using limits that are defined by physical possibility for each site. For the precipitation gauge bucket weight data, this also included the removal of data jumps related to gauge servicing (bucket emptying and/or charging). Anything missed or flagged by the automated quality control process was then examined and managed manually.

Quality control of the snow survey data was largely completed at the time of digitization when the field observation sheets were transferred into a spreadsheet. Snow depth data were plotted and examined for outliers, which were generally from misreading the snow rod or incorrectly transcribing the observation in the field. Outliers were removed and not included in the site mean and standard deviation. The same was done for the density samples.

3.3 Precipitation post-processing and amalgamation

The quality controlled 1-minute bucket weight data from the precipitation gauges were first smoothed using a Gaussian filter with a 4-minute running window. This filter smoothed spikes in the time series resulting from mechanical or electrical noise. The time series were then zeroed to the start of the season and further filtered using a revised version of the Brute Force precipitation filter developed by Environment and Climate Change Canada Climate Research Division, introduced by Pan et al. (2016), and henceforth called the Neutral Aggregating Filter (NAF). Although Pan et al. briefly describes the filter, the NAF and some subsequent improvements are described in more detail here.

NAF is an automated method to remove noise from cumulative precipitation time series by iteratively balancing positive and negative noise until all changes below a user defined threshold, $\Delta^*$, are eliminated. $\Delta^*$ is typically set to 0.05 to 0.2 mm, depending on the gauge precision; 0.05 mm is used for this study. The algorithm removes random noise and accounts for diurnal oscillations in the bucket weight signal (likely resulting from differential heating of the sensors; Duchon, 2008) but does not account for negative drift, which means that it will not perform well if the time series has significant periods with evaporative losses from the accumulating gauge bucket. The significance of the error depends on the user's tolerance of the loss relative to the total precipitation but could exceed 10% depending on the effectiveness of the servicing measures to reduce evaporation from the bucket.

The algorithm is conceptually simple: all non-zero changes in interval precipitation, $\Delta(t)$, with values below $\Delta^*$ are transferred to neighboring periods with positive changes. The processing is done iteratively, beginning with the minimum non-zero $\Delta(t)$ value. The results from the algorithm are neutral, that is, they preserve the total cumulative precipitation from the raw bucket weight time series, which is why evaporation results in an estimation error.

Following the quality control described above, the NAF algorithm processing steps are as follows:

1. The change in interval precipitation, $\Delta(t)$, is computed as the difference between the bucket weights in consecutive periods (1-minute in the case of the SPICE data) in the cumulative time series. If a gap in the bucket weight data exists, the difference is computed across the gap. Because $\Delta(t)$ contains noise, it may be positive, negative or zero.
2. All non-zero $\Delta(t)$ values that are less than $\Delta^*$ are identified and ranked from smallest (most negative) to largest, the minimum of which becomes $\Delta(i)$.
3. All points with $\Delta(t)>0$ are also identified.
4. From the point $\Delta(i)$, the nearest point with $\Delta(t)>0$ is found (either before or after $\Delta(i)$), which will become $\Delta(j)$.
5. $\Delta(i)$ is added to $\Delta(j)$ and $\Delta(i)$ is set to zero. If two equidistant $\Delta(t)>0$ points are found (before and after $\Delta(i)$), then $\Delta(i)$ is split between the two by adding $0.5^*\Delta(i)$ to each. Note that the resulting value for $\Delta(j)$ may remain positive, or it may become zero or negative.
6. Steps (2) to (5) are repeated for the revised time series from (5), first re-ranking the non-zero $\Delta(t)<\Delta^*$ points (steps (2) and (3)), then repeating steps (4) to (5) with the new lowest $\Delta(i)$ value. This is repeated until there are no remaining points with $\Delta(t)<\Delta^*$. This becomes the filtered interval precipitation, with all non-zero values of $\Delta(t)$ greater than or equal to $\Delta^*$.
7. The cumulative precipitation time series is calculated as the cumulative sum of the non-missing values from (6).
8. The results from (7) are inspected by plotting the difference between the filtered and original cumulative precipitation time series. Periods where the differences diverge significantly from zero (e.g. by more than 1 mm) are an indication of an anomaly in the cumulative time series, likely the result of evaporative losses.

After (8) above, if there are no large divergences in the cumulative time series, the NAF cumulative time series can be accepted without further processing. If periods with significant divergences are found, the time series should be visually inspected to identify the cause, and if possible, known problems should be eliminated from the time series. As indicated above, the most common issue is negative drift resulting from evaporation from the bucket contents, in which case the NAF flattens out periods of negative drift between precipitation events. This typically occurs several times within the time series. The result is an underestimation of the precipitation amount at the start and end of each flattened, diverging period, and sometimes the elimination of small precipitation events during the period (see Fig. 5). There can also be spurious excursions over shorter intervals that have no apparent cause and need removal.

Typically, raw precipitation data from accumulating gauges, such as those presented here, have enough inherent evaporation or spurious excursions to create accumulating errors in seasonal precipitation as high as 10% of the total, following the NAF process. At this time, there is no automated procedure to satisfactorily reduce this error. For this reason, a supervised process for adjusting the cumulative time series for evaporation and other spurious data was developed. This process uses NAF as the first guess and allows the user to interactively select the end points of diverging periods (identified in Step 8 above) or the spurious events, and the algorithm effectively adjusts the "flattening" of the cumulative time series. This supervised process, called NAF-S, effectively reduces the impact of evaporation but does require some user subjectivity to identify the end points.

Figure 5 shows an example of the NAF (red) and NAF-S (black) post-processing of the raw precipitation gauge bucket weight following data quality control and the application of a Gaussian filter (blue) on an excerpt of real bucket weight data obtained at XBK in October of 2015. Note how the NAF (using a $\Delta$* of 0.05 mm in this example) effectively removes the high frequency and diurnal noise in the raw bucket weight data but fails to account for the evaporation signal from October 5-9, thereby affecting the quantity of the precipitation event that begins on October 11. NAF-S adjusts and compensates for this error, resulting in an increase in the precipitation estimate as compared to the NAF first-guess.

Following NAF-S on the SPICE data, the 1-minute time series were resampled to 30-minute intervals. The difference between the 30-minute bucket weights are the 30-minute precipitation amounts reported in this data set.

3.4 Missing meteorological and precipitation data

The missing data values in the SK SPICE dataset were set to the numeric value of -999 and generally occurred when the instrument malfunctioned, the data failed to collect (e.g. logger or power outage), or were removed during the quality control process. No gap filling of the meteorological data was performed. During data outages, the precipitation gauges continued to accumulate precipitation whether or not the data were recorded, and thereby preserved the accumulated precipitation measurement during the outage. Although the user can't determine when this precipitation occurred during the outage, the total amount is known via the total change in bucket weight. Data

in the record, regardless of the source, were flagged with a numeric value of 1 in the Flag column to indicate that more than one third of the 1-minute values were missing from the aggregation. In the case of precipitation, if the 30 minute reported value represented a period longer than 30 minutes, the Flag column was also 1 and the period length can be determined by counting the number of previously missing 30-minute periods.

### 3.5 Wind undercatch

The precipitation data published here were not adjusted for wind undercatch. However, this data set includes all of the ancillary data required to perform adjustments using various published techniques and transfer functions (i.e. Wolff et al., 2015, Kochendorfer et al., 2017b, Smith, 2009). The data flags were included to assist the user in

making an adjustment for wind. Precipitation data with a Flag=1 (see above) were not adjusted because the wind and temperature conditions during the actual precipitation event were unknown.

### 4 Precipitation summaries

Table 2 shows the seasonal accumulations of precipitation for both Bratt's Lake and Caribou Creek and for the various gauge configurations. Note that the seasonal totals are for 1 October through 30 April, unless noted otherwise. Several seasonal accumulations are abbreviated due to data availability beginning later in the year. These are indicated as incomplete with an (I) in the table and the beginning of the accumulation period is noted in the footnote under the table. The corresponding accumulated precipitation time series are shown in Fig. 6.

Although the intent of this paper is not to present an intercomparison of the gauge configuration catch efficiencies nor the precipitation differences between the sites, there are several points that can be made about these observations. Although it is difficult to ascertain due to incomplete seasons, the winter precipitation (as measured by the DFAR) is generally greater at CCR than it is at XBK. The October-April 2013/2014 accumulations in Table 2

are an example of this, and is consistent with the climate normals indicated in Table 1. Table 1 also shows that the average gauge height wind speed at XBK (4.4 m s$^{-1}$) is almost double the average for CCR (2.6 m s$^{-1}$), which explains the relative catch efficiency of the Geonor SA (as compared to the DFAR) at each site. The seasonal Geonor SA catch at CCR ranges from 80% to 82% while the catch for the same configuration at XBK is considerably less, varying with season from 59% to 69%. As anticipated, the sheltered Geonor Bush gauge at CCR

has a high relative catch as compared to the DFAR (92% to 97%) but always has a lower catch than the DFAR, which is contrary to the catch of the manual bush shielded gauge observed at Valdai (Yang et al., 1993; Yang, 2014). These results are also shown in the time series (Fig. 6) with the wind undercatch of the Geonor SA at XBK quite prominent as demonstrated by the often rapid deviation between the accumulated DFAR and Geonor SA precipitation at this site. This is in contrast with the Geonor SA at CCR that only deviates substantially from the

DFAR during very windy events. The Geonor Bush accumulated precipitation at CCR very closely tracks with the accumulated DFAR precipitation.

Figure 7 shows the mean transect SWE observations from Caribou Creek, calculated as the product of the mean transect snow depth (n=50) and the mean transect density (n=5), shown in units of mm of water equivalent (w.e.). The error due to the covariance between snow depth and density (Steppuhn, 1976) is usually small in snowpacks shallower than 80 cm (Pomeroy and Gray, 1995) and therefore is not included in Fig. 7. Calculated covariance is included in the published dataset and is generally under 2% of the mean SWE. Although 2015/2016 is absent, the highest SWE was measured in 2013/2014 (127 mm w.e. on Feb. 26). SWE is much more variable in 2014/2015 and 2016/2017, with 2014/2015 having the lowest peak of the three seasons shown. The peak SWE in 2016/2017 (99 mm w.e.) was observed late in the season (Mar. 23) as a result of a large snowfall event which began on March 5 (Fig. 6 bottom right).

## 5 Applications

The precipitation data collected at the Bratt's Lake and Caribou Creek sites during the SPICE intercomparison period (2013/2014 and 2014/2015) were contributed to the WMO-SPICE intercomparison and used to develop the SPICE transfer functions (Kochendorfer et al., 2017b, 2018). The snow survey data over the same period were used as the reference for assessing the performance of an automated SWE sensor (Smith et al., 2017). With the continuation of the data collection at these sites, the 2015/2016 and 2016/2017 (and beyond) data are used for an independent assessment of the SPICE transfer functions, providing data from both the reference gauge configuration (DFAR) and a test gauge configuration (Geonor SA). Figure 8, as an example, shows the unadjusted (solid black) and adjusted Geonor SA (solid red and blue; using the SPICE Eq. 3 and Eq. 4 transfer functions from Kochendorfer et al., 2017b) accumulated time series of precipitation at Caribou Creek (Fig. 8a) and Bratt's Lake (Fig. 8b) for the 2016/2017 winter as compared to the accumulated DFAR (dashed black) for the same period. Preliminary results from Caribou Creek (Fig. 8a) suggest that both of the SPICE transfer functions (Eq. 3 which incorporates air temperature and Eq. 4 which does not) over-adjust the winter precipitation at this site by approximately 8%. Alternatively, the preliminary results from Bratt's Lake (Fig. 8b) suggest that both transfer functions under-adjust the winter precipitation at this site by nearly 25% in this example. Although post-SPICE validation of the SPICE transfer functions is ongoing, results from Caribou Creek and Bratt's Lake suggest that the SPICE transfer functions tend to over-adjust at less windy sites and under-adjust at more windy sites, consistent with the results shown by Kochendorfer et al. (2017b). Extrapolation of the transfer function performance to sites without a DFAR can only be speculative.

Within the CCRN program, Pan et al. (2016) recently carried out precipitation bias adjustments at several research sites in the CCRN domain, however Bratt's Lake and Caribou Creek were not included. That analysis used a transfer function derived from a single test site to adjust precipitation measured in the much wider network of CCRN stations, resulting in an unknown uncertainty in the application. The application of the SPICE transfer functions is also not without uncertainty (as shown in Fig. 8) but one would expect that transfer functions based on multiple sites and combined data should be more widely applicable and therefore used for future precipitation data adjustments in

cold regions. This Saskatchewan SPICE and post-SPICE data set has, and will continue to be a valuable asset for both testing and refining precipitation adjustment methodologies.

## 6 Data and code availability

The Saskatchewan SPICE data from the winters of 2013/2014 through 2016/2017 can be found on the Government of Canada Open Data portal at: https://doi.org/10.18164/63773b5b-5529-4b1e-9150-10acb84d59f0. This includes the 30-minute precipitation and ancillary data (temperature and wind speed) from both Caribou Creek and Bratt's Lake and the bi-weekly or monthly snow survey summaries from Caribou Creek. The metadata, also found at the

above link, describes the data format and summarizes the information in this manuscript. It can be downloaded in both French and English. The MATLAB code package, including the NAF and NAF-S scripts and documentation, are available as a supplement to this manuscript or can be obtained by contacting the corresponding author.

## 7 Summary

The Bratt's Lake and Caribou Creek Saskatchewan SPICE data collected by ECCC during the winters (October through April) of 2013/2014 to 2016/2017 includes the WMO DFAR as a solid precipitation reference measurement, the single Alter Geonor T-200B (which is the configuration most commonly used in the MSC climate network), a bush shielded Geonor T-200B (at CCR only as a proxy for a bush measurement as at Valdai, Russia), wind speed at gauge height, and air temperature. Although these data are not all of the data collected during and

after WMO-SPICE at CCR and XBK, they do include the core precipitation and ancillary measurements. Available on the Government of Canada Open Data Portal, these data have been, and will continue to be used for instrument intercomparisons and validation of precipitation gauge transfer functions. It will be a useful data set for NWP and Hydrological model validation and remote sensing ground-truthing, with the intercomparison sites and infrastructure available for future in-situ intercomparison projects.

## 8 Author contribution

Craig Smith was responsible for overseeing the collection, quality control, archiving and distribution of the Saskatchewan SPICE data and managed the research activities at the Saskatchewan SPICE sites. Dr. Daqing Yang provided site management support and contributed expertise for the measurement of solid precipitation. Amber Ross was responsible for the precipitation data quality control and data post-processing. Dr. Alan Barr developed and

coded the NAF and NAF-S precipitation post-processing procedures documented in this manuscript.

## 9 Competing interests

The authors declare that they have no conflict of interest.

**10 Special issue statement**

This article is part of the special issue "Water, ecosystem, cryosphere, and climate data from the interior of Western Canada and other cold regions". It is not associated with a conference.

**11 Acknowledgements**

The authors would like to thank all of the students and staff of the Climate Research Division and the Watershed Hydrology and Ecology Research Division of ECCC who contributed to the success of the SPICE field projects in Saskatchewan, including Cuyler Onclin, Bruce Cole, Lauren Arnold, Scott Wood, Stephnie Watson, and Emma Wattie. We appreciate the contributions of Dr. Michael Earle (ECCC) and Audrey Reverdin (MeteoSwiss) in the

development of the WMO-SPICE quality control procedures and for the project leadership of Rodica Nitu (ECCC). We would especially like to extend our gratitude to the reviewers, Drs. Tedd Hogg and John Kochendorfer and the anonymous reviewers, who provided their valuable time to help us improve this manuscript.

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

**Table 1:** Saskatchewan SPICE site locations (Latitude, Longitude, and Elevation), mean annual air temperature ($T_{air}$), mean annual total precipitation (P), and mean wind speed at gauge height ($U_{gh}$).

| Site (Abbreviation) | Lat. | Lon. | Elev. | Mean $T_{air}$* | Mean P* | Mean $U_{gh}$** |
|---|---|---|---|---|---|---|
| Bratt's Lake (XBK) | 50.200° | -104.711° | 585 m | 3.1 °C | 389.7 mm | 4.4 m s$^{-1}$ |
| Caribou Creek (CCR) | 53.945° | -104.649° | 519 m | 0.9 °C | 427.3 mm | 2.6 m s$^{-1}$ |

*From the 1981-2010 Environment Canada Climate Normals at the nearest long term climate station (Regina Airport for Bratt's Lake and Nipawin Airport for Caribou Creek)

**Mean for the 2013-2015 SPICE period at the site, approx. 2 m above the ground

**Table 2:** Seasonal totals of precipitation (October through April, where available) and the relative catch (%) for the Geonor SA and Geonor Bush as compared to the DFAR. Incomplete seasonal totals (I) are usually due to precipitation data starting later than October 1 (see footnotes).

| | XBK | | CCR | | |
|---|---|---|---|---|---|
| Year | DFAR(mm) | Geonor SA(mm) | DFAR(mm) | Geonor Bush(mm) | Geonor SA(mm) |
| 2013/2014 | 170.2 | 100.7 (59%) | 279.6 | 259.0 (93%) | 224.1 (80%) |
| 2014/2015 | 141.3(I[1]) | 78.4(I[1]) (56%) | 106.3(I[2]) | 98.7(I[2]) (93%) | 86.8 (I[2]) (82%) |
| 2015/2016 | 48.1(I[3]) | 33.2(I[3]) (69%) | 189.3 | 174.3 (92%) | No data |
| 2016/2017 | 168.8 | 104.4 (62%) | 164.2(I[4]) | 159.6(I[4]) (97%) | 134.4(I[4]) (82%) |

[1]begins 11 Oct 2014; [2]begins 4 Dec 2014; [3]begins 1 Dec 2015; [4]begins 9 Nov 2016

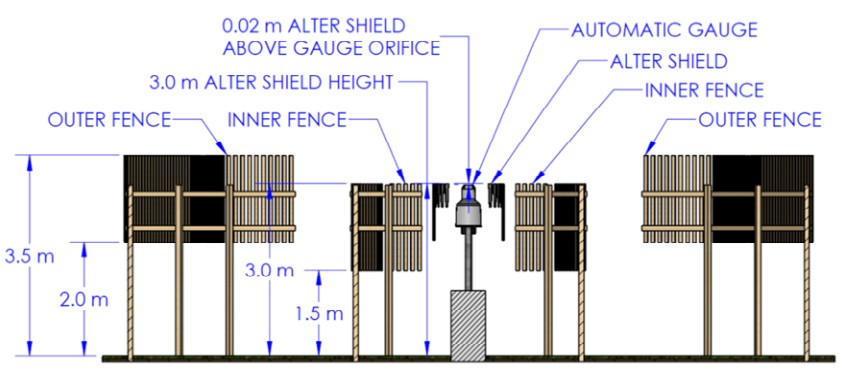 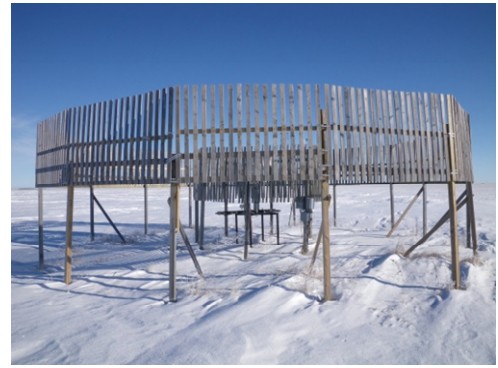

**Figure 1:** Conceptual diagram (left; Nitu & Roulet, 2016; diagram courtesy of Jeff Hoover, Environment and Climate Change Canada) and photo (right; Bratt's Lake) of the WMO DFAR.

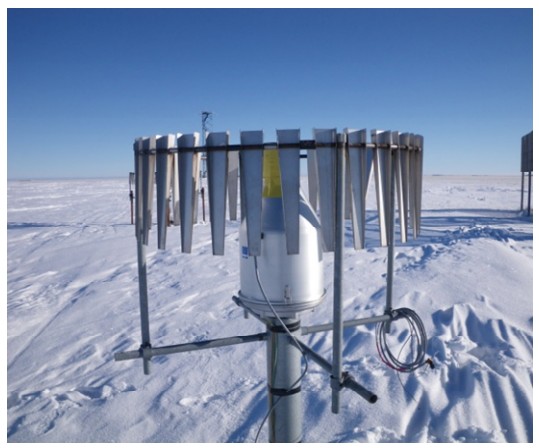 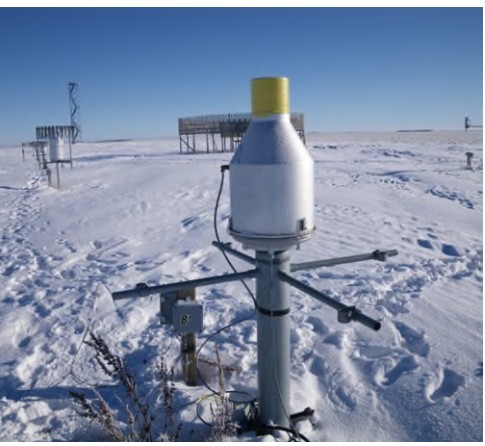

**Figure 2:** The SPICE Alter shielded (left) and unshielded (right) Geonor T-200B gauge pair at Bratt's Lake.

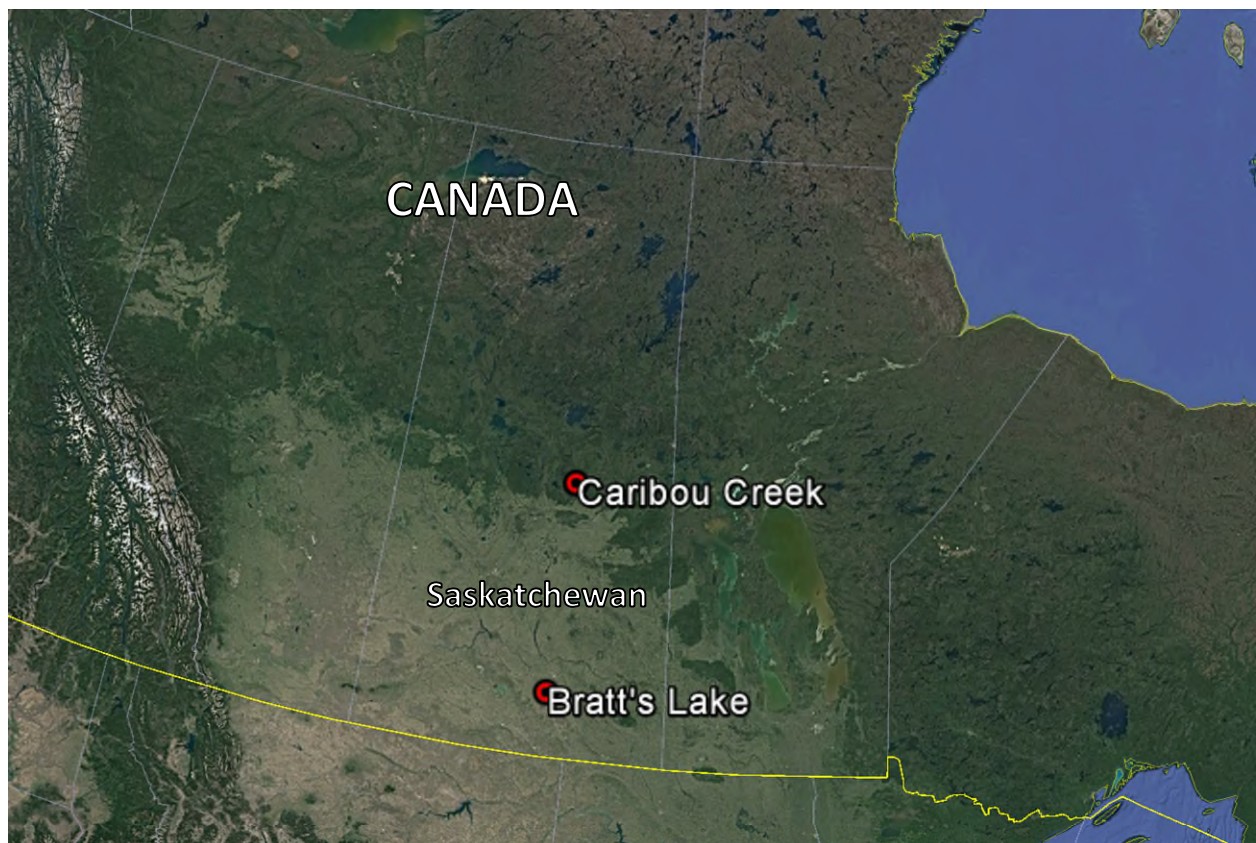

**Figure 3:** Location of the Caribou Creek and Bratt's Lake SK sites in western Canada (base map obtained from Google earth; Data SIO, U.S. Navy, GEBCO ©2018 Google; Image Landsat/Copernicus)

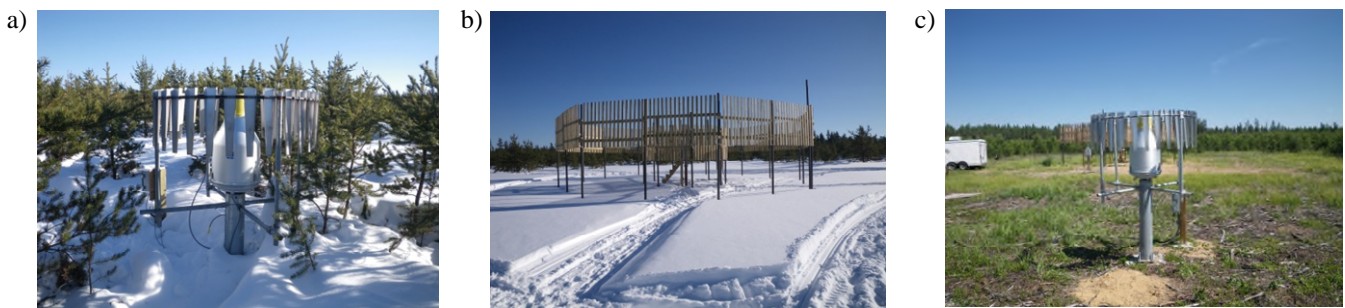

**Figure 4:** Precipitation gauge installations at the Caribou Creek SPICE site: a) Bush shielded Geonor T-200B with Alter shield, b) DFAR with Geonor T-200B in clearing and c) Alter shielded Geonor T-200B in clearing.

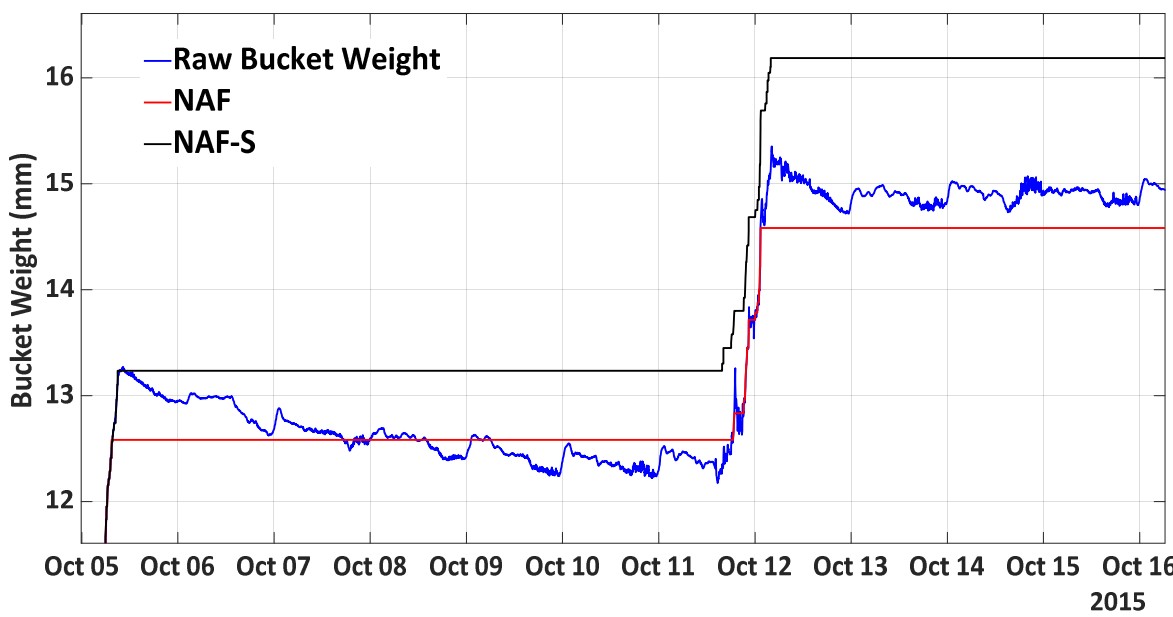

**Figure 5:** NAF (red) and NAF-S (black) precipitation data time series data processing example as compared to the Gaussian filtered raw bucket weight (blue). Data excerpt is from an actual precipitation time series observed at XBK in October 2015.

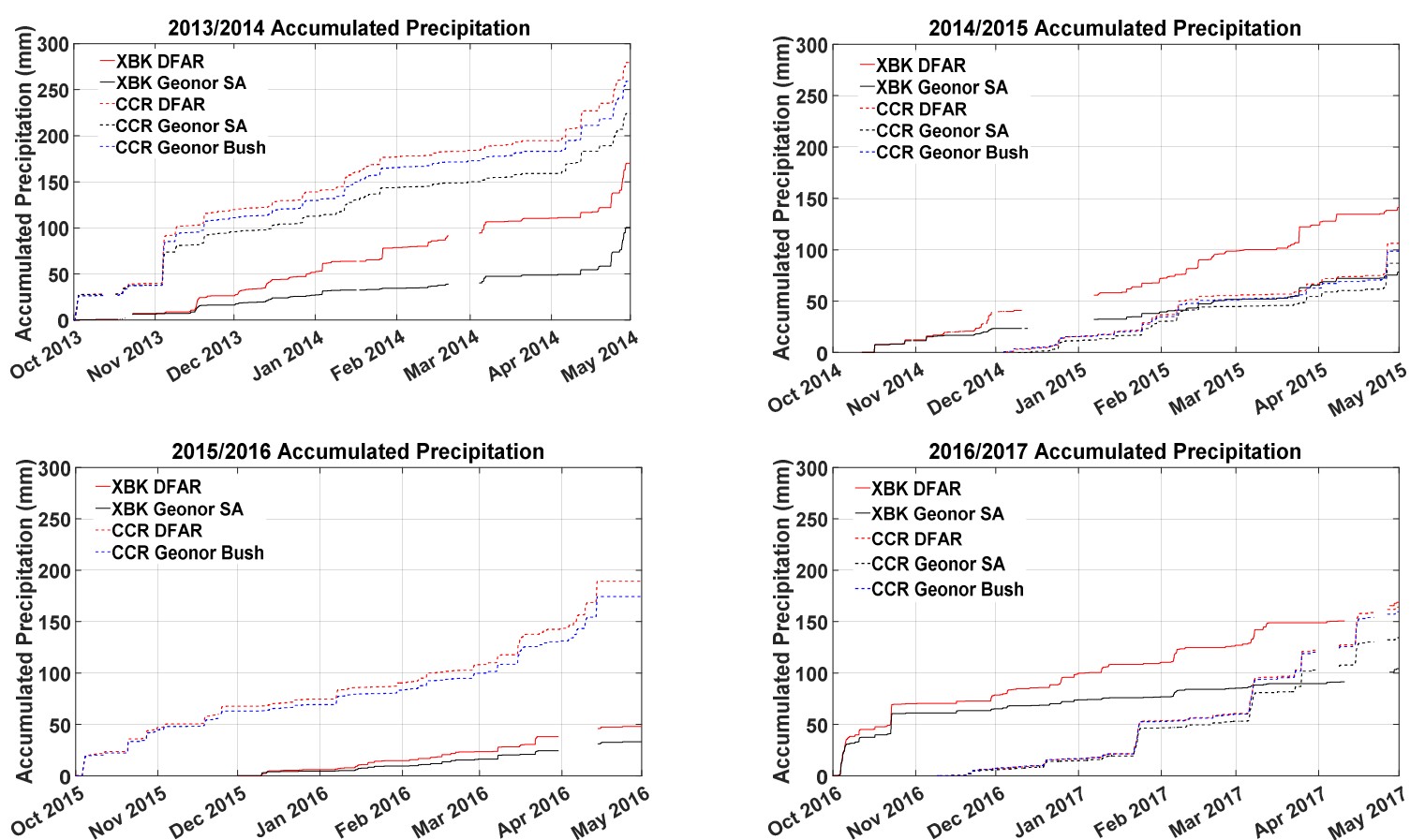

**Figure 6:** Seasonal time series of accumulated precipitation for the various gauge configurations at Bratt's Lake and Caribou Creek. Note that although the accumulation season is from 1 October through 30 April, not all time series start at the beginning of the season due to gauge or site issues (see Table 2).

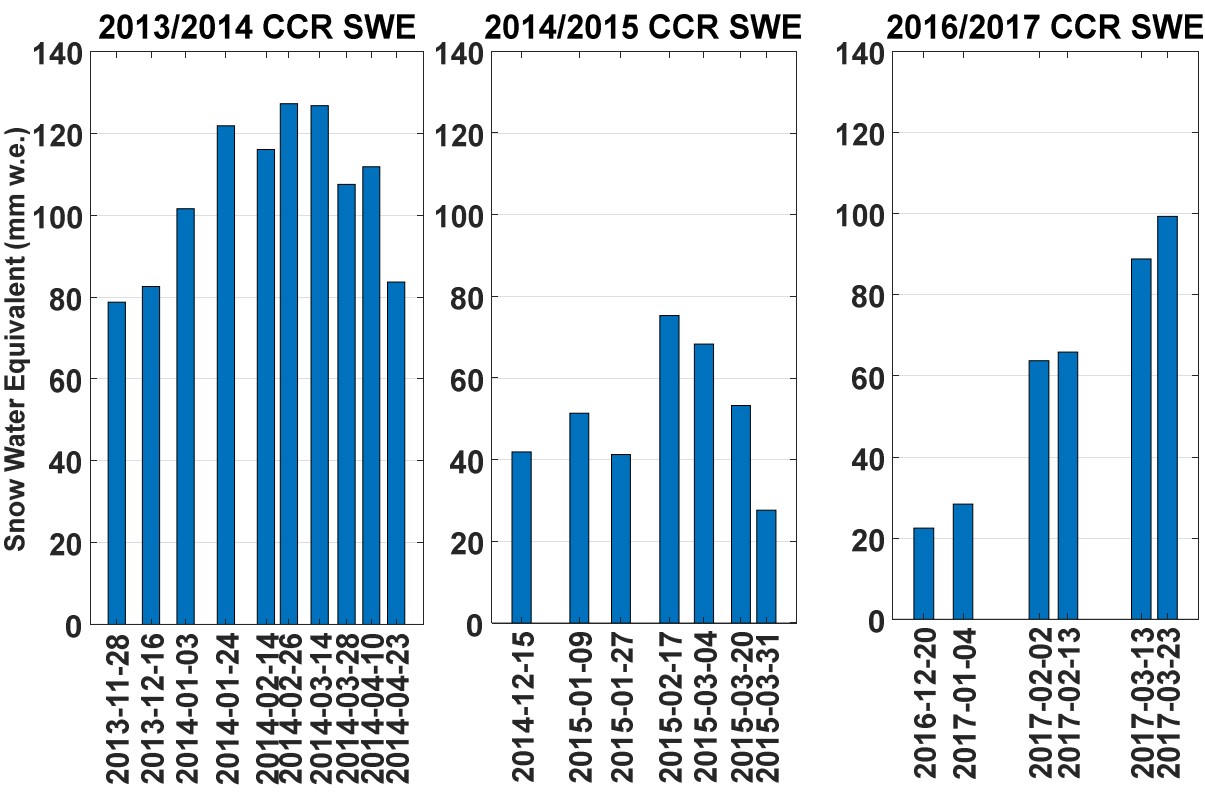

**Figure 7:** Caribou Creek SWE SWE observations by date for 2013/2014, 2014/2015, and 2016/2017.

a)

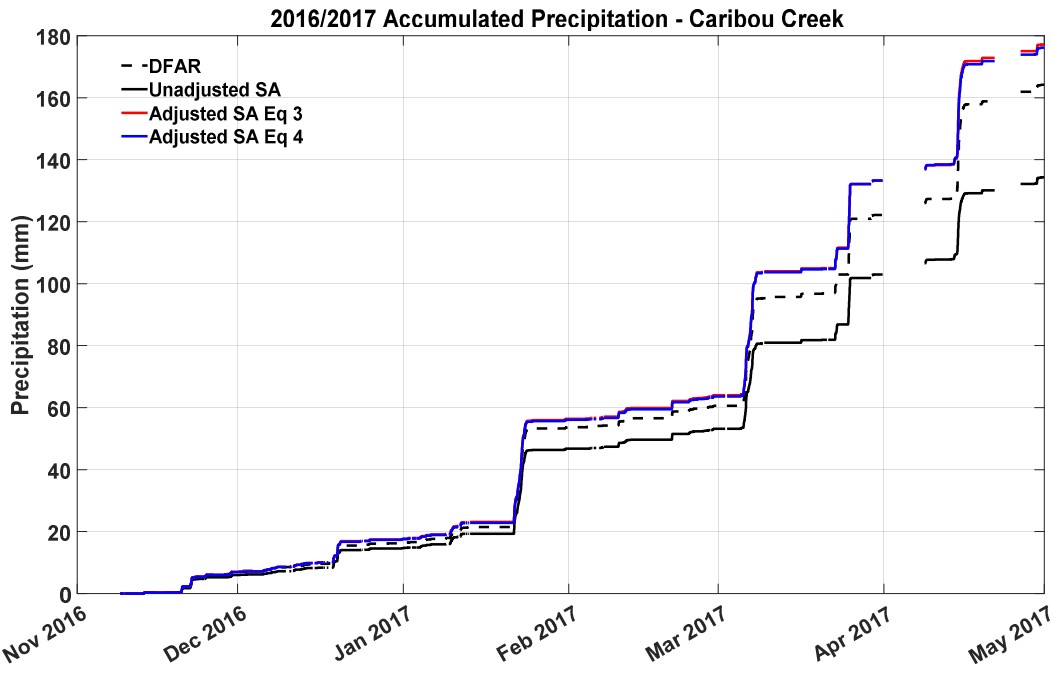

b)

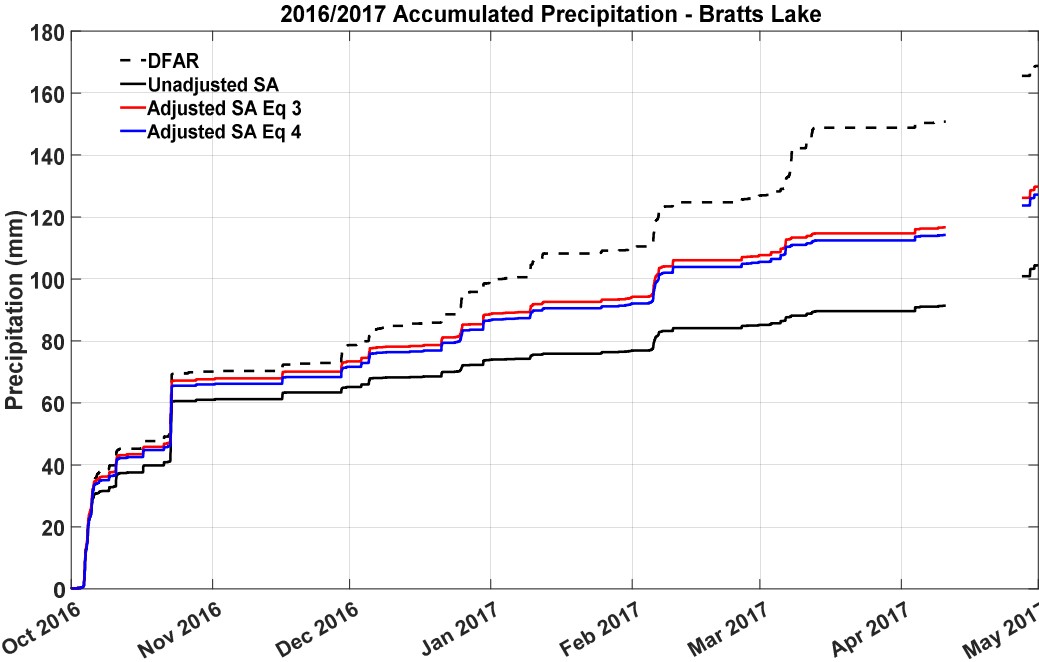

**Figure 8:** 2016/2017 winter accumulated precipitation time series from a) Caribou Creek and b) Bratt's Lake showing the unadjusted single Alter Geonor T-200B (solid black), the adjusted single Alter Geonor T-200B (read and blue solid; via the SPICE Eq. 3 and Eq. 4 transfer functions from Kochendorfer et al. 2017b), and the DFAR (dashed black).