# Peer review of "The Environment and Climate Change Canada solid precipitation intercomparison data from Bratt's Lake and Caribou Creek, Saskatchewan"

_Earth System Science Data, 2018_

## Referee Comment (RC1) · Anonymous Referee #1 · 31 Oct 2018

The treatment of the accumulated precipitation data is well explained, but the actual software used is not given. If a program not written by the authors was used, it should be mentioned.

p. 2, line 30 - add "air" before "temperature"

line 25 the phrase "and reported here" is a bit confusing - it sounds like you are referring to the document, rather than the data set. Suggest you delete it, or change to "as specified in the data set"

[Figure]

line 26 change "reported here" to "included in the data set"

p. 4. line 6 Is the Bratt's Lake site grassed? From the online description, it appears to be cropped.

line 8 "initially" - was it changed? If not, this can be deleted.

line 11 "am" should be "an"

line 20 "best illustrated" If the heaters aren't visible in any other images (they don't appear to be), change to "seen"

line 27 insert "they" before "generally"

p. 5, line 7 add "were" between "data" and "retrieved"

line 14 "realistic" is an odd word. Perhaps this could be explained.

line 22 change "or" to "and"

p. 6, line 6 The missing data values, not necessarily the flags, are set to -999

Figure 3 This image appears to be taken from Google Earth, but is not credited. Is it permitted to publish Google Earth images?

---

## Short Comment (SC1) · 5 Nov 2018

General comment:

This paper provides an overview of a hydrological experiment in Saskatchewan, Canada, together with a description of the data sets that are being made publicly available on a Canadian federal government web site (Environment and Climate Change Canada). The experiment was aimed at the comparison of winter precipitation measurements (primarily snow) using alternative instrumentation and wind-shelter sys-

tems. Measurements were made over four successive winters at two contrasting sites in the boreal forest (regenerating jack pine) and on the open (treeless) prairie. The context for the research is that conventional designs for measuring precipitation have been found to substantively underestimate the input of moisture by snow, especially in open and windy environments. Overall, the paper is clearly written, and I appreciate the efforts of the authors to facilitate the sharing and archiving of data from this important experiment. In my specific points below, I have offered some comments and suggestions that are mainly aimed at 1) making the work more accessible to an international readership and 2) increasing the usefulness of the presented results.

Specific points on the manuscript:

P1 Abstract: For an international readership, I think it would be helpful to start the abstract with one or two sentences highlighting the rationale for conducting these experiments, i.e., the problem of snow undercatch by conventional precipitation guages in windy environments, and the need for specialized instrumentation systems and/or models to address this problem. Also, the abstract seems rather detailed in its description of methods, whereas I see no reporting of results or their implications. Having said this, I recognize that I am not familiar with what is normally expected for papers such as this that are primarily focused on the publication of data sets.

P3 L30: I recommend defining acronymns such as "SWE" (Snow Water Equivalent) when they are first mentioned.

P6 L10: Awkward sentence structure. I suggest re-wording to something like "Although this precipitation amount cannot be distributed to specific times during the outage, it is retained.."

P6 L20, "It is strongly suggested that precipitation data with a Flag=1 (see above) not be adjusted as..": I would just say "Precipitation data with a Flag=1 (see above) were not adjusted because..".

P6, section "4 Precipitation summaries": This reads like a Results section but it seems rather cursory and brief (see also my comments below on Table 2 and on Figures 5 and 6).

P7 L6-9: The results reported here are interesting and appear to be important, i.e., the transfer functions overestimated winter precipitation at the forested site but give underestimates at the prairie site. Would it be worth adding a sentence or two on the broader implications of these results? Or alternatively, is there another paper from this study that could be cited for readers who may wish to explore these issues further?

P12 Table 2: For the CCR site in 2014/2015 and 2016/2017, the reported measurements of seasonal precipitation are of little value for making comparisons because they cover different time periods. It would be more useful to report on precipitation totals over the seasonal period each year when all three systems were operating. The authors might also consider reporting the seasonal totals for the Geonor Bush and Geonor SA systems as a percentage of the totals obtained using DFAR.

P15 Figure 5: I expect that some of the accumulated precipitation was rain rather than snow, especially in October and April-May. Given that the problem of precipitation undercatch is (likely) greater for snow than for rain events, the authors might consider adding a horizontal bar or vertical dotted lines showing the winter period when the predominant source of precipitation was snow.

P 16 Figure 6: This figure shows SWE measurements for one of the two sites (Caribou Creek). Is there any comparable data from the other site, Bratt's Lake? Also, it is not clear to me how these data were used in the analysis of snow catch efficiency for the different precipitation measurement systems.

Comments on the external link containing the data set

I successfully accessed the web page containing the data set, at: http://data.ec.gc.ca/data/climate/scientificknowledge/saskatchewan-solidprecipitation-inter-comparison-experiment-spice-data/

The web page provides a good overview of the experiment, including the general location of the two sites and the measurements included in the data set.

A map server shows the general region of Saskatchewan where the two sites are located; however, the actual site locations are not shown on the map.

On this web page, there is also a hyperlink to the Metadata, but when I attempted to access it I received a message saying "Connection refused". Thus, I am not able to provide feedback on this material.

However, under the "Resources" section of the web page, there are two MS Word files (in English and French) that provide the at least some of the information I would expect to find in a metadata file, including a description of the study, with specific site locations, methods of quality control, and a description of each variable in the data set.

The data files also include 16 comma-delimited ".dat" files containing half-hourly measurements of precipitation and seasonally accumulated precipitation using the alternative instrumentation systems, along with wind speed and air temperature. As a test, I imported one of these ".dat" files into MS Excel and the formatting appears to be sound and easy to understand. In addition, there are two MS Word files (in English and French) that provide tabular summaries of the snow survey data sets from the Caribou Creek site, along with some graphics showing snow densities and snow water equivalents.

---

## Short Comment (SC2) · 5 Nov 2018

Please note that I am one of the invited referees for this paper. The "Short comment" that I posted above should read "Referee comment". Although I indicated "Anonymous Referee #2", I am okay with including my name with my review.

---

## Referee Comment (RC2) · J. Kochendorfer (Referee) · 9 Nov 2018

**General Comments:**

The manuscript describes measurements recorded at two Canadian precipitation intercomparison sites. The history of the sites, along with a description of the measurements recorded at the sites is described. The relationship between the measurements and WMO-SPICE is also discussed, and the important fact that there are more data
5 available from these sites than was included in WMO-SPICE is made clear. Large uncertainties in solid precipitation measurements are accurately described as the motivation for both WMO-SPICE and the dataset that the manuscript describes. In addition, the testing of WMO-SPICE transfer functions serves as a nice example of using the measurements to help advance precipitation research.

My primary criticism of the paper is that it does not include many significant and new scientific developments.
10 Other than the testing of transfer functions shown in Figure 7, the content of the manuscript is mainly descriptive and historic. I admit however that this may be an unfair criticism, as the stated scope of the journal is to promote "the reuse of high-quality datasets" rather perhaps than present new science. The present manuscript does appear to have less analysis and new results than some of the recent ESSD manuscripts I looked over, but this impression may be due in part to my own unfamiliarity with the focus of the other ESSD manuscripts I found. In addition, ESSD
15 does appear to publish some other similarly descriptive manuscripts. So this criticism should be taken with a grain of salt. I am not suggesting that the manuscript be rejected on this account, but it would certainly strengthen to paper to include more new and interesting results.

The manuscript is generally well written and presented, but some mainly small editorial and technical recommendations are made below.

20 **Specific Comments:**

P. 1, l. 16. Replace "issues" with a more specific word. And rewrite or delete, "It is also fortunate that".

P. 1, l. 26. Reword, "and reduced to monthly" – it is awkward as written.

P.1, l. 35-36. Rewrite "The systematic bias issues in the measurement of snowfall" as "Snowfall measurement biases".

25 P. 2, l. 3. Citation error – Kochendorfer et al. didn't publish anything relevant in 2016.

P. 2, l. 36. Change "represents" to, "are". Not sure how to resolve this, but I think "data" are plural, and "data set" may be singular, so it isn't clear if "are" or "is" would be more appropriate – make it consistent. Consider deleting "represents a high quality precipitation data set" all together, as it doesn't add much.

P. 3, l. 1. Why does the data "represent" a contribution? Rewrite this sentence more succinctly.

30 P. 3, l. 16. Change, "tree heights *averaging* about 2 to 3 m", to, "tree heights *of* about 2 to 3 m." Change "opportunistic" to "opportune". Arguably this sentence and the one before it should be changed to past tense as well. The tense changes from present to past somewhat arbitrarily throughout this paragraph. It is probably just easier to stick with past tense. For more general guidance: https://www.nature.com/scitable/topicpage/effective-writing-13815989

35 P. 3, l. 26-28. Clarify what is meant by "not reported" – is it excluded from the dataset or just the manuscript? Maybe there is a clearer way of describing this? Or if it is included in the dataset "not reported" can probably just be removed.

P. 3, l. 32. Remove the word "site" from "site SWE".

P. 4, l. 1. Change "crossing" to, "and crossed".

P. 4, l. 11. Typo – change "*am* R.M. Young…" to "*an* R.M. Young…".

P. 4, l. 13. Clarify if the Stevenson screen was fan aspirated or naturally aspirated.

P. 4, l. 26 - 27. Change to, "although *they* generally kept…".

P. 5, l. 10 p 6, l. 21. Change Section 3.3 – 3.5 to mainly past tense.

P. 5, l. 31 – 35. I like this approach, and the description is clear. But unless you plan on publishing it elsewhere, describe the BFU filter in more detail. For example, what types of numerical techniques were used, and is this software available for others to use? This manuscript would be a good opportunity to publish the method, and it would also help give the manuscript a bit more substance.

Figure 5. The font size used throughout the figure is too small. It is difficult to read.

P. 6, l. 34. Change, "were a contribution" to, "were contributed".

P. 7, l. 9. How was ~30% calculated? Based on the figure it looks like prior to the missing data, the SA was only under-adjusted by ~25% ((150 mm – 115 mm)/150 mm).

---

## Author Comment (AC1) · 2 Jan 2019

RC: The treatment of the accumulated precipitation data is well explained, but the actual software used is not given. If a program not written by the authors was used, it should be mentioned.

AC: The program used to process the data was written in MATLAB by co-author Alan Barr and previously used (and briefly described) in Pan et al. (2015). Although the testing and intercomparison of this filtering algorithm will be the subject of a separate publication and is somewhat out of scope here, the description of the algorithm in Section 3.3 has been expanded substantially and more thoroughly than the brief description contained in the original Pan et al. reference.

Action: The filtering algorithm used on this precipitation data will be better described in the manuscript and it will be made clear that the algorithm was written in-house by the co-authors.

p. 2, line 30 - add "air" before "temperature"

Action: done

line 25 the phrase "and reported here" is a bit confusing - it sounds like you are referring to the document, rather than the data set. Suggest you delete it, or change to "as specified in the data set"

Action: removed this sentence to avoid confusion

line 26 change "reported here" to "included in the data set"

Action: revised as suggested

p. 4. line 6 Is the Bratt's Lake site grassed? From the online description, it appears to be cropped.

Action: added the following sentence to clarify: "The observation site is mown grass surrounded by agricultural crops."

line 8 "initially" - was it changed? If not, this can be deleted.

Action: revised as suggested

line 11 "am" should be "an"

Action: revised as suggested

line 20 "best illustrated" If the heaters aren't visible in any other images (they don't appear to be), change to "seen"

Action: revised as suggested

line 27 insert "they" before "generally"

Action: revised as suggested

p. 5, line 7 add "were" between "data" and "retrieved"

Action: revised as suggested

line 14 "realistic" is an odd word. Perhaps this could be explained.

AC: "realistic" means physically possible for the site.

Action: This sentence was changed to "This is an automated process which removes out-of-range outliers and data jumps, the thresholds for which are set using limits that are defined by physical possibility for each site."

line 22 change "or" to "and"

Action: revised as suggested

p. 6, line 6 The missing data values, not necessarily the flags, are set to -999

AC: agreed

Action: revised

Figure 3 This image appears to be taken from Google Earth, but is not credited. Is it permitted to publish Google Earth images?

AC: Yes, the background image was obtained from Google Earth but the attributes were cropped when optimizing the figure. The Google Earth attributes guidelines are described at https://www.google.com/permissions/geoguidelines/attr-guide/.

Action: The attributes of this base image have been reintroduced in the figure caption, as per the online guidelines.

**T. Hogg (Referee #2)**
ted.hogg@canada.ca

General comment:
This paper provides an overview of a hydrological experiment in Saskatchewan,
Canada, together with a description of the data sets that are being made publicly available
on a Canadian federal government web site (Environment and Climate Change
Canada). The experiment was aimed at the comparison of winter precipitation measurements
(primarily snow) using alternative instrumentation and wind-shelter systems. Measurements were made over
four successive winters at two contrasting sites in the boreal forest (regenerating jack pine) and on the open
(treeless) prairie. The context for the research is that conventional designs for measuring precipitation have
been found to substantively underestimate the input of moisture by snow, especially in open and windy
environments. Overall, the paper is clearly written, and I appreciate the efforts of the authors to facilitate the
sharing and archiving of data from this important experiment. In my specific points below, I have offered some
comments and suggestions that are mainly aimed at 1) making the work more accessible to an international
readership and 2) increasing the usefulness of the presented results.

AC: Thank you Dr. Hogg for the insightful review. I just wanted to clarify that the intent of this data paper
submission to the "Water, ecosystem, cryosphere, and climate data from the interior of Western Canada and
other cold regions" special issue was not to present new or unique intercomparisons amongst gauge types and
windshield configurations, although intercomparisons are included to help characterize the dataset and give the
reader/data user context on the impact of shielding in these dissimilar environments. The intent, rather, is to
document and highlight the winter precipitation data set collected at these two sites for future use by land-
surface, climate, or hydrological modellers, remote sensing validation, etc.  With that in mind, I hope that we
have satisfactorily addressed your comments and suggestions and have improved the manuscript.

Specific points on the manuscript:
P1 Abstract: For an international readership, I think it would be helpful to start the
abstract with one or two sentences highlighting the rationale for conducting these experiments,
i.e., the problem of snow undercatch by conventional precipitation guages
in windy environments, and the need for specialized instrumentation systems and/or
models to address this problem. Also, the abstract seems rather detailed in its description
of methods, whereas I see no reporting of results or their implications. Having said
this, I recognize that I am not familiar with what is normally expected for papers such
as this that are primarily focused on the publication of data sets.

AC: We agree that there should be a statement in the abstract that frames both the issue with gauge
undercatch of snow and the experiments designed to resolve this issue in the observations. As you note, the
abstract (and the paper) are heavily focused on the collection, QC, and processing methods rather than
intercomparison results. This was the intent. More comprehensive intercomparison results from the WMO-
SPICE project (which included these two sites and 6 additional international sites) can be found in the SPICE
special issue of HESS (https://www.hydrol-earth-syst-sci.net/special_issue400_78.html), many of which
are referenced in the manuscript.

Action: The abstract has been revised. The first two sentences are now: "Prior to the beginning of the World
Meteorological Organization's (WMO) Solid Precipitation Inter-Comparison Experiment (SPICE, 2013-2015),
two precipitation measurement intercomparison sites were established in Saskatchewan to help assess the
systematic bias in the automated gauge measurement of solid precipitation and the impact of wind on the
undercatch of snowfall. Caribou Creek, located in the southern Boreal forest, and Bratt's Lake, located in the
southern plains, are a contribution to the international SPICE project but also to examine national and regional
issues in measuring solid precipitation, including assessment of precipitation gauges and wind shield
configurations commonly used in Canadian monitoring networks."

P3 L30: I recommend defining acronymns such as "SWE" (Snow Water Equivalent)
when they are first mentioned.

Action: acronym has now been defined when it is first used

P6 L10: Awkward sentence structure. I suggest re-wording to something like "Although this precipitation amount cannot be distributed to specific times during the outage, it is retained.."

Action: This sentence is replaced with: Although the user can't determine when this precipitation occurred during the outage, the total amount is known via the total change in bucket weight." I think this is much clearer.

P6 L20, "It is strongly suggested that precipitation data with a Flag=1 (see above) not be adjusted as..": I would just say "Precipitation data with a Flag=1 (see above) were not adjusted because..".

Action: revised as suggested

P6, section "4 Precipitation summaries": This reads like a Results section but it seems rather cursory and brief (see also my comments below on Table 2 and on Figures 5 and 6).

AC: The "results" reported here are somewhat brief as this was not intended to be a true intercomparison of the sites and gauges. However, to improve the readability of the manuscript, we have added some additional commentary on the measurements and an intercomparison in Section 4.

P7 L6-9: The results reported here are interesting and appear to be important, i.e., the transfer functions overestimated winter precipitation at the forested site but give underestimates at the prairie site. Would it be worth adding a sentence or two on the broader implications of these results? Or alternatively, is there another paper from this study that could be cited for readers who may wish to explore these issues further?

AC: We agree with the reviewer. Based on ongoing work with all 8 international SPICE sites, we can see that the transfer functions generally over-adjust at non-windy sites and under-adjust at windy sites. The implications on the data are mostly obvious, although we can only speculate on how the transfer functions will work at sites that do not have a reference for intercomparison.

Action: We added a sentence to comment on the implications of the behaviour of the transfer functions at these two sites. This now states: "Although post-SPICE validation of the SPICE transfer functions is ongoing, results from Caribou Creek and Bratt's Lake suggest that the SPICE transfer functions tend to over-adjust at less windy sites and under-adjust at more windy sites, consistent with the results shown by Kochendorfer et al. (2017b). Extrapolation of the transfer function performance to sites without a DFAR can only be speculative."

P12 Table 2: For the CCR site in 2014/2015 and 2016/2017, the reported measurements of seasonal precipitation are of little value for making comparisons because they cover different time periods. It would be more useful to report on precipitation totals over the seasonal period each year when all three systems were operating. The authors might also consider reporting the seasonal totals for the Geonor Bush and Geonor SA systems as a percentage of the totals obtained using DFAR.

AC: Agreed.

Action: The CCR Geonor Bush total for 2014/2015 was re-calculated to begin 4 Dec 2014 to be consistent with the other two gauges. The same was done for 2016/2017 at CCR, where the season was adjusted to begin 9 Nov 2016. Figure 5 was updated accordingly to match the season lengths shown in Table 2. We also added the percentage of total catch as compared to the DFAR for each season and for each gauge.

P15 Figure 5: I expect that some of the accumulated precipitation was rain rather than snow, especially in October and April-May. Given that the problem of precipitation undercatch is (likely) greater for snow than for rain events, the authors might consider adding a horizontal bar or vertical dotted lines showing the winter period when the predominant source of precipitation was snow.

AC: In practice, this is a difficult thing to do, especially as we've chosen to plot both sites on the same graph for each winter season. There can also be significant rain events mid-winter. In the end, I'm not really sure that delineating the season in this way would be that useful. It is, however, outlined in the methodology how temperature is used to make precipitation phase decisions for adjustment purposes. The user can do the same using the published temperature data.

P 16 Figure 6: This figure shows SWE measurements for one of the two sites (Caribou Creek). Is there any comparable data from the other site, Bratt's Lake? Also, it is not clear to me how these data were used in the analysis of snow catch efficiency for the different precipitation measurement systems.

AC: SWE measurements were not made at the Bratt's Lake site. This data was not used in any way to assess the catch efficiency of the gauges as this was not the intent of the paper. Simply, this data is a winter precipitation product collected during this CCRN study period and is provided and documented for the future use of the reader of this special issue.

Comments on the external link containing the data set
I successfully accessed the web page containing the data set, at:
http://data.ec.gc.ca/data/climate/scientificknowledge/saskatchewan-solid- precipitation-inter-comparison-experiment-spice-data/

The web page provides a good overview of the experiment, including the general location of the two sites and the measurements included in the data set.
A map server shows the general region of Saskatchewan where the two sites are located; however, the actual site locations are not shown on the map.

AC: The GC Open Data Portal map server only allows the delineation of area and not, oddly enough, point data points. The GPS locations of the site are included however, and the map in the manuscript pinpoints the site locations in SK.

On this web page, there is also a hyperlink to the Metadata, but when I attempted to access it I received a message saying "Connection refused". Thus, I am not able to provide feedback on this material.

However, under the "Resources" section of the web page, there are two MS Word files (in English and French) that provide the at least some of the information I would expect to find in a metadata file, including a description of the study, with specific site locations, methods of quality control, and a description of each variable in the data set.

AC: I will inquire with the database administrators why that link is broken but the metadata, as you point out, is listed below the link under "Resources" in both of French and English.

The data files also include 16 comma-delimited ".dat" files containing half-hourly measurements of precipitation and seasonally accumulated precipitation using the alternative instrumentation systems, along with wind speed and air temperature. As a test, I imported one of these ".dat" files into MS Excel and the formatting appears to be sound and easy to understand. In addition, there are two MS Word files (in English and French) that provide tabular summaries of the snow survey data sets from the Caribou Creek site, along with some graphics showing snow densities and snow water equivalents.

**J. Kochendorfer (Referee #3)**
john.kochendorfer@noaa.gov

**General Comments:**

The manuscript describes measurements recorded at two Canadian precipitation intercomparison sites. The history of the sites, along with a description of the measurements recorded at the sites is described. The relationship between the measurements and WMO-SPICE is also discussed, and the important fact that there are more data available from these sites than was included in WMO-SPICE is made clear. Large uncertainties in solid precipitation measurements are accurately described as the motivation for both WMO-SPICE and the dataset that the manuscript describes. In addition, the testing of WMO-SPICE transfer functions serves as a nice example of using the measurements to help advance precipitation research.

My primary criticism of the paper is that it does not include many significant and new scientific developments. Other than the testing of transfer functions shown in Figure 7, the content of the manuscript is mainly descriptive and historic. I admit however that this may be an unfair criticism, as the stated scope of the journal is to promote "the reuse of high-quality datasets" rather perhaps than present new science. The present manuscript does appear to have less analysis and new results than some of the recent ESSD manuscripts I looked over, but this impression may be due in part to my own unfamiliarity with the focus of the other ESSD manuscripts I found. In addition, ESSD does appear to publish some other similarly descriptive manuscripts. So this criticism should be taken with a grain of salt. I am not suggesting that the manuscript be rejected on this account, but it would certainly strengthen to paper to include more new and interesting results.

The manuscript is generally well written and presented, but some mainly small editorial and technical recommendations are made below.

AC: Thank you Dr. Kochendorfer for your comments and suggestions. As pointed out to Referee #2, the intent of this data paper was not to present new scientific results, but rather to document and publish a data set that could have potential applications for other cold region research projects relevant to readers of the ESSD special issue "Water, ecosystem, cryosphere, and climate data from the interior of Western Canada and other cold regions".  This data will be used by the authors (and potentially others) for analysis and results presented in other scientific papers. However, in response to other comments, we have added more substance to the "Precipitation summaries" section to discuss more the intercomparison of the data and the implications of the observed biases.

**Specific Comments:**

P. 1, l. 16. Replace "issues" with a more specific word. And rewrite or delete, "It is also fortunate that".

Action: both of these sentences have been revised. They now read "…but also to examine national and regional issues in measuring solid precipitation, including regional assessment of wind bias in precipitation gauges and wind shield configurations commonly used in Canadian monitoring networks. Overlapping with WMO-SPICE, the Changing Cold Regions Network (CCRN) Special Observation and Analysis Period (SOAP) occurred from 2014 to 2015 involving other enhanced observations and cold regions research projects in the same geographical domain as the Saskatchewan SPICE sites."

P. 1, l. 26. Reword, "and reduced to monthly" – it is awkward as written.

Action: revised as suggested

P.1, l. 35-36. Rewrite "The systematic bias issues in the measurement of snowfall" as "Snowfall measurement biases".

Action: revised as suggested

P. 2, l. 3. Citation error – Kochendorfer et al. didn't publish anything relevant in 2016.

Action: removed this citation

P. 2, l. 36. Change "represents" to, "are". Not sure how to resolve this, but I think "data" are plural, and "data set" may be singular, so it isn't clear if "are" or "is" would be more appropriate – make it consistent. Consider deleting "represents a high quality precipitation data set" all together, as it doesn't add much.

Action: revised as suggested

P. 3, l. 1. Why does the data "represent" a contribution? Rewrite this sentence more succinctly.

Action: revised as suggested

P. 3, l. 16. Change, "tree heights *averaging* about 2 to 3 m", to, "tree heights *of* about 2 to 3 m." Change "opportunistic" to "opportune". Arguably this sentence and the one before it should be changed to past tense as well. The tense changes from present to past somewhat arbitrarily throughout this paragraph. It is probably just easier to stick with past tense. For more general guidance: https://www.nature.com/scitable/topicpage/effective-writing-13815989

AC: Does it really make sense to talk about an existing site in past tense? "The site is located…" vs. "The site was located…" I left the tense in these sentences unrevised but will defer to the technical editor.

P. 3, l. 26-28. Clarify what is meant by "not reported" – is it excluded from the dataset or just the manuscript? Maybe there is a clearer way of describing this? Or if it is included in the dataset "not reported" can probably just be removed.

AC: This sentence was already revised as suggested by another reviewer

P. 3, l. 32. Remove the word "site" from "site SWE".

Action: revised as suggested

P. 4, l. 1. Change "crossing" to, "and crossed".

Action: revised as suggested

P. 4, l. 11. Typo – change "*am* R.M. Young…" to "*an* R.M. Young…".

Action: revised as suggested

P. 4, l. 13. Clarify if the Stevenson screen was fan aspirated or naturally aspirated.

AC: The screen is fan aspirated.

Action: sentence is clarified

P. 4, l. 26 - 27. Change to, "although *they* generally kept…".

Action: revised as suggested

P. 5, l. 10 p 6, l. 21. Change Section 3.3 – 3.5 to mainly past tense.

Action: revised tense where required

P. 5, l. 31 – 35. I like this approach, and the description is clear. But unless you plan on publishing it elsewhere, describe the BFU filter in more detail. For example, what types of numerical techniques were used, and is this software available for others to use? This manuscript would be a good opportunity to publish the method, and it would also help give the manuscript a bit more substance.

AC: This is a very good suggestion, and one made my another reviewer. We will include a much more comprehensive description of the filtering technique in Section 3.3, including a name change (from Brute Force to Instrument Noise Aggregating Filter or INAF), a visual example of the processing, and upload the documented MATLAB code containing the algorithm as a supplement. Although testing and intercomparison of filtering techniques is out of scope of this paper, we intend to do a thorough review of the performance of this filter and others commonly used operationally, and publish these results in a separate manuscript.

Figure 5. The font size used throughout the figure is too small. It is difficult to read.

AC: Agreed. The font size will be increased before final submission

P. 6, l. 34. Change, "were a contribution" to, "were contributed".

Action: revised tense where required

P. 7, l. 9. How was ~30% calculated? Based on the figure it looks like prior to the missing data, the SA was only under-adjusted by ~25% ((150 mm – 115 mm)/150 mm).

AC: This 30% value was based on the multiple years of data that were adjusted and not just the 2016/2017 adjustment shown in the example.

Action: This is clarified in the manuscript.

---

## Author Response (AR2)

**The following is the response to Report #1 submitted by Anonymous Referee #1. All points have been addressed and we thank the referee for their suggestions.**

**RC:** The error is located at
Page 8, Line 32
5  "..SWE measurements, calculated as the product of the mean transect snow depth(n=50) and the mean transect density (n=5)"

This is incorrect. The mean transect SWE (not the SWE measurement) is a product of the mean depth and the mean density, with the addition of the covariance between the depth
10  and density. This is very well established in the literature,
see Faria, D.A., Pomeroy, J.W. and Essery, R.L.H., 2000. Effect of covariance between ablation and snow water equivalent on depletion of snow-covered area in a forest. Hydrological Processes, 14(15), pp.2683-2695.

**AC:** The text has been updated to state the following: "Figure 7 shows the mean transect SWE
15  observations from Caribou Creek SWE, calculated as the product of the mean transect snow depth (n=50) and the mean transect density (n=5), shown in units of mm of water equivalent (w.e.). The error due to the covariance between snow depth and density (Steppuhn, 1976) is usually small in snowpacks shallower than 80 cm (Pomeroy and Gray, 1995) and therefore is not included in Fig. 7. Calculated covariance is included in the published dataset and is generally under 2% of the mean
20  SWE." The data summary has been updated to include the calculation of the covariance.

RC: Page 2 Line 10
"The DFAR configuration consists of the same large
octagonal double wind fence used by the WMO Double Fence Intercomparison Reference"
25  Is the the very same fence, or just the same type of fence? If the former, no change is required.

**AC:** No, it is not the "very same" fence but a fence built using the same specifications as the WMO DFIR. The text has been updated to reflect this.

**RC:** Line 15 "a catch efficiency of 94%, 92% and 90%"
30  Should probably be "catch efficiencies'

**AC:** agreed, corrected

**RC:** Page 5 Line 36
"diurnal oscillations in the bucket weight signal"
35  This is very interesting - is this a well known issue in the precipitation measurement community?
It would be useful to have some speculation on the
cause, i.e. is it an issue with the electronics, or is it some
type of condensation?

**AC:** In the Geonor gauge, the exact source of this "noise" is speculative within the user community.
40  The most popular theory is that is a result of differential heating of the sensors. This has been clarified in the manuscript with a brief description and a reference.

**RC:** Page 6 Line 5
"positive or larger changes"
I'm not sure what this means.

5   **AC:** This should just be "…positive changes." and has been fixed in the manuscript

**RC:** Page 7 Line 28
"-999"

The use of quotation marks around -999 implies that it is a character string
in the file, which may be confusing to people using the data.

I would suggest rewriting the sentence to indicate that this is a numerical value, e.g.
15   Missing data in the SK SPICE dataset were set to a value of -999...

  **AC:** Corrected

RC: Note that sometimes the precipitation flag is "1" and sometimes it is 1,
which is also confusing

20   **AC:** Corrected

[revised manuscript text omitted]